# Antibiotic Activity Screened by the Rheology of *S. aureus* Cultures

**Raquel Portela [1], Filipe Valcovo [1], Pedro L. Almeida [2,3], Rita G. Sobral [1] and Catarina R. Leal [2,3,*]**

[1] UCIBIO@REQUIMTE, Faculdade de Ciências e Tecnologia, Universidade Nova de Lisboa, 2829-516 Caparica, Portugal; rp.portela@campus.fct.unl.pt (R.P.); f.valcovo@campus.fct.unl.pt (F.V.); rgs@fct.unl.pt (R.G.S.)

[2] Área Departamental de Física, ISEL—Instituto Superior de Engenharia de Lisboa, Instituto Politécnico de Lisboa, 1959-007 Lisboa, Portugal; palmeida@adf.isel.pt

[3] CENIMAT/I3N, Departamento de Ciência dos Materiais, Faculdade Ciências e Tecnologia, Universidade Nova de Lisboa, 2829-516 Caparica, Portugal

\* Correspondence: cleal@adf.isel.pt; Tel.: +351-21-831-7135

**Abstract:** Multidrug resistant bacteria are one of the most serious public health threats nowadays. How bacteria, as a population, react to the presence of antibiotics is of major importance to the outcome of the chosen treatment. In this study we addressed the impact of oxacillin, a β-lactam, the most clinically relevant class of antibiotics, in the viscosity profile of the methicillin resistant *Staphylococcus aureus* (MRSA) strain COL. In the first approach, the antibiotic was added, at concentrations under the minimum inhibitory concentration (sub-MIC), to the culture of *S. aureus* and steady-state shear flow curves were obtained for discrete time points during the bacterial growth, with and without the presence of the antibiotic, showing distinct viscosity progress over time. The different behaviors obtained led us to test the impact of the sub-inhibitory concentration and a concentration that inhibited growth. In the second approach, the viscosity growth curves were measured at a constant shear rate of 10 s$^{-1}$, over time. The obtained rheological behaviors revealed distinctive characteristics associated to the presence of each concentration of the tested antibiotic. These results bring new insights to the bacteria response to a well-known bacteriolytic antibiotic.

**Keywords:** rheology; MRSA; *S. aureus*; antibiotics; oxacillin; bactericidal

## 1. Introduction

Bacterial multidrug resistance is the root of the antibiotic crisis that humankind is beginning to face and that will become disastrous in the very near future. In view of the increasing shortage of available therapies that are efficient against bacterial strains that have adapted and developed numerous resistance mechanisms, new strategies are needed. Treatment failure is often the consequence of sub-populations of cells that become resistant or tolerant to the antibiotic pressure. Many parameters influence the outcome, such as presence of other microorganisms, the host immune system or nutrient limitations. Another important factor, often neglected, is the shear stress conditions to which bacteria are submitted, inside and outside the host. To develop new antibacterial strategies, knowledge must be obtained on how the bacteria respond to the presence of antibiotics and under shear stress, as a complex population both as biofilms and in the planktonic state.

It is already well established that the viscoelasticity of biofilms, which critically depends on their structure and composition, plays a major role in their protective effect against mechanical and chemical challenges [1]. In this regard, one can find several works in the literature concerning the characterization of the mechanical behavior of bacterial biofilms, although most are implemented over solid biofilms as they occur in real situations [2–4], to study the adhesion properties to surfaces.

A recent study compared the impact of having complex medium tryptic soy broth supplemented by glucose and NaCl (TGN) or artificial sputum medium (ASM), that mimics cystic fibrosis sputum (CFS), as the nutrient support for biofilm formation, in the response of *S. aureus* to several antibiotics. The authors claimed that the higher elastic properties of ASM, very similar to the ones of CFS, contribute actively towards the observed outcome, that all antibiotics were drastically less efficient in ASM than in TGN towards clearing of *S. aureus* [5].

In our recent works, the mechanical behavior of *S. aureus* planktonic cultures was accessed by biological methods in conjunction with rheological and rheo-imaging techniques [6–10]. When subjected to a shear flow, the bacterial cultures of *S. aureus* disclosed an intricate rheological behavior without a counterpart in biological procedures characterization. More specifically, in steady shear flow, the viscosity value was enhanced during the exponential phase and reverted throughout the late phase to a value similar to the original one, complemented with the equilibrium of the cell population, exhibiting shear rate dependence [7]. The observed decrease in the viscosity is mainly caused by cell deposition [10]. In oscillatory flow, both elastic and viscous moduli presented power-law dependencies whose exponents are functions of the bacteria growth phase and can be concomitant with a soft glassy material response [11]. These behaviors were enclosed in a microscopic model that considers the development of a dynamic web-like structure, where particular aggregation phenomena might arise, according to the growth phase and bacterial density. By merging optical density assays and dry weight determination procedures, recent studies brought new indications confirming that the bacterial aggregation patterns that arise throughout growth, under shear, cannot be attributed uniquely to a cell population density dependence [9].

In this study we addressed the impact of the β-lactam antibiotic oxacillin (Oxa) on the early methicillin resistant *S. aureus* (MRSA) strain COL [12]. MRSA strains emerged in the 1960s, after the introduction of β-lactamase-resistant β-lactam antibiotics, the class of antibiotics most used in the clinic, that inhibit bacterial cell wall synthesis by binding irreversibly to the transpeptidase domain of penicillin-binding proteins (PBPs) [13]. PBP proteins catalyze the last steps of peptidoglycan biosynthesis, in the external side of the membrane, being responsible for the high cross-linking level of this cell wall macromolecule. The MRSA strains, that are intrinsically resistant to the β-lactam mode of action, emerged through the acquisition of a mobile genetic element, the staphylococcal cassette chromosome *mec* (SCC*mec* cassette), the genetic element carrying the β-lactam resistance gene *mecA*, from other bacterial species [14]. The acquisition of an SCC*mec* cassette occurred in numerous independent events, resulting in distinct MRSA clonal lineages, but all SCC*mec* cassettes contain the *mecA* gene, the central element of resistance. The *mecA* gene encodes an extra PBP for PBP2A, with low affinity for virtually all β-lactam antibiotics, resulting in a lower efficiency of acylation that takes over cell wall synthesis in the presence of β-lactam antibiotics [15]. As for other β-lactam antibiotics, the mode of action of oxacillin is described as being bactericidal, since by inactivating PBPs, the cell wall weakens and the cell eventually lyses [16]. The MRSA strain COL presents a high minimum inhibitory concentration (MIC) value of 400 µg/mL [17]. The MIC corresponds to the lowest concentration of an antibacterial drug that inhibits the growth of bacteria.

The antibiotic oxacillin was added to the cultures of the *S. aureus* strain, in different concentrations and different rheological experimental approaches were applied. Namely, steady-state flow curves and viscosity growth curve measurements. Moreover, the viscosity growth curves were obtained at a constant shear rate value over time, compatible with human physiological values [18]. The obtained rheological behaviors revealed distinctive characteristics associated with the presence of the tested antibiotic. These differences are justified by the antibiotic bacteriolytic mode of action.

## 2. Materials and Methods

### 2.1. Bacterial Strain and Growth Conditions

The methicillin resistant *Staphylococcus aureus* (MRSA) strain COL, a gram-positive human pathogen [19] was used. The strain was grown overnight in tryptic soy agar medium (TSA) plates at 37 °C. A single colony of the strain was inoculated in 5 mL of fresh tryptic soy broth (TSB) and grown overnight at 37 °C at 180 rpm to promote the aeration of the culture.

Cultures were grown at 37 °C with aeration in an orbital shaker (180 rpm) and the optical density at 620 nm ($OD_{620\,nm}$) was monitored over time using a spectrophotometer (Ultrospec 2100 pro).

To determine the most appropriate concentrations of antibiotic to study, growth curves were performed in the presence of oxacillin at lower and higher concentrations than the MIC value (200, 400, 800 and 1600 μg/mL) added to the culture at 180 min of growth.

The influence of the antibiotic in the growth process of the culture was assessed considering two approaches regarding rheological characterization: A, Addition of antibiotic at a specific time of growth, namely at t = 300 min, to the culture while still in the incubator, followed by the rheological measurement; B, addition of antibiotic at specific time points of growth during the rheological measurement. In both approaches *S. aureus* cultures were set by inoculating fresh TSB medium with the starting cultures to obtain an initial $OD_{620\,nm} = 0.005$.

*Approach A*:

At ~300 min of growth, 200 μg/mL of oxacillin was added to the culture in the flask. A sample was taken to continue growth in the rheometer. After the addition of the antibiotic the cultures remained in the incubator with the $OD_{620\,nm}$ monitored. At time points, 1, 2 and 3 h of incubation, samples of the cultures were taken in triplicate to determine the cell viable counts (CFU/mL) by plating serial dilutions of the bacterial cultures on TSA medium without antibiotic. For the same time points, a calibrated volume of sample was collected and observed by optical microscopy using an Olympus BX50 microscope with an Olympus SC30 camera and AnalySIS getIT 5.1 image software. For each aliquot, an average of 10 photos were taken randomly. This optical analysis was also applied to the concentration of oxacillin of 800 μg/mL.

*Approach B*:

The cultures were loaded on the rheometer at an $OD_{620\,nm}$ of 0.005 and grown at 37 °C under shear. At ~340 min of growth, oxacillin was added to a final concentration of 200 and 800 μg/mL. This procedure was carefully performed to minimize the interference in the running measurement, such as to prevent contact with the rotating upper geometry and to assure the dispersion of the antibiotic inside the sample.

### 2.2. Rheology

Rheological measurements were performed in a controlled stress rotational rheometer Bohlin Gemini HRnano, and different geometries were used to perform two types of assays:

(i)　flow curve: a steel cone and plate geometry, with diameter 40 mm, angle 2° and gap 55 μm for the steady-state shear flow tests, which were performed at 20 °C. Each step was acquired assuring that a minimum of 1500 units of deformation was imposed to the sample. Using a CP geometry with a very small gap, sample ejection, due to centrifuge effect, did not occur within the range of shear rate values considered;

(ii)　viscosity growth curve: a steel plate-plate geometry, with diameter 40 mm and gap of 2000 μm (to ensure a good signal) for the steady shear viscosity measurement, imposing a constant shear rate of 10 s$^{-1}$, during time and at 37 °C to allow optimal growth conditions.

In approach B tests, the addition of the antibiotic was performed using a micropipette with a tip diameter of 0.5 mm which is 4 times smaller than the gap used in the measurement. For control purposes we have previously used an acrylic PP geometry, to follow the addition of a dye to the sample,

in the same volume as the antibiotic, which gave us an insight on how the dispersion occurs, and no peripherical droplet accumulation was verified.

Assays were performed in triplicate over fresh cultures and a solvent trap was used to minimize evaporation.

## 3. Results and Discussion

### 3.1. Antibiotic Effect on S. aureus Growth

Previous studies showed that the oxacillin MIC value of strain COL in solid medium was 400 µg/mL [17]. To study the rheological effect of the challenge of oxacillin on an MRSA strain, we first determined the most appropriate concentrations of antibiotic to use, lower and higher than the MIC value (200, 400, 800 and 1600 µg/mL). The optical density of a COL culture, in function of the added antibiotic concentration, was measured over growth and, as expected, the growth rate of the population challenged with increasing concentrations of oxacillin, decreased accordingly, see Figure 1a.

We chose the concentration of 200 µg/mL, corresponding to half the MIC value to perform a first set of experiments, steady state shear flow tests. This concentration of oxacillin resulted in a slower growth rate, during the exponential phase, but the cell division process was clearly still active, as observed by optical density monitoring, see Figure 1a. For further rheological studies, we also chose the concentration of 800 µg/mL, corresponding to twice the MIC value. This concentration of oxacillin resulted in an arrest in growth, approximately 100 min after antibiotic addition (see Figure 1b), followed by a decrease in the optical density of the culture, suggesting arrest of cell division associated with cell lysis.

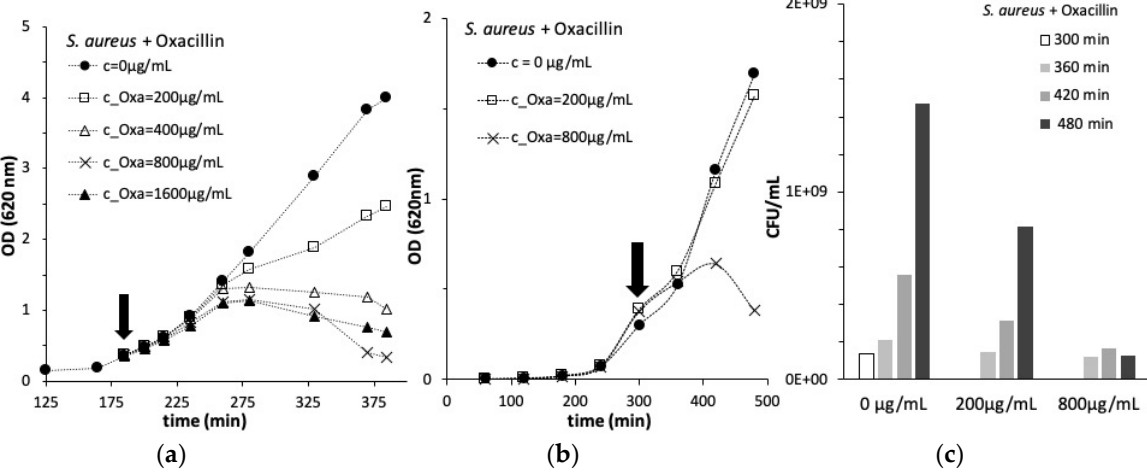

**Figure 1.** (**a**) Growth curves of *S. aureus* strain COL growth monitored by measuring the optical density (OD$_{620\,nm}$) over time. *S. aureus* was grown in the absence and in the presence of different concentrations of oxacillin (Oxa), at 200, 400, 800 and 1600 µg/mL added at t = 180 min (representative curves). The arrow represents the time of addition of the antibiotic to the culture. (**b**) Growth curves of *S. aureus* strain COL grown in the absence and in the presence of different concentrations of antibiotics: oxacillin (Oxa) at 200 and 800 µg/mL added at 300 min (representative curves). The arrow represents the time of addition of the antibiotic to the culture. (**c**) Colony forming units (CFU/mL), at specific growth times: 300, 360, 420 and 480 min after the addition of oxacillin. All the measurements were performed at 37 °C.

### 3.2. Steady State Shear Flow Assays

Steady state shear flow tests were performed at 20 °C (to minimize bacteria growth during the assay) and allowance made for the characterization of the viscosity of the cultures, in the absence and in presence of a sub-inhibitory concentration of oxacillin (200 µg/mL) in function of the shear rate.

In general, the flow curves revealed a shear-thinning behavior followed by a shear-thickening behavior for the *S. aureus* cultures, with and without the presence of antibiotics. In Figure 2a the flow curves for the *S. aureus* cultures with the addition of oxacillin are plotted for different growth times (for *S.aureus* flow curves please see Figure 2 in [6]). For samples collected at the beginning of the $OD_{620\,nm}$ growth curve (see Figure 1b), the viscosity appears to be smaller than the culture medium viscosity, Figure 2a. Only for intermediate and longer growth times, 395 min and 485 min, the viscosity shows higher values than the culture medium viscosity, for shear rate values lower than 300 $s^{-1}$, in the shear-thinning region. For higher shear rate values, above 600 $s^{-1}$, all the flow curves tend into the same curve with almost the same slope of the culture medium.

The representation of the viscosity values obtained from the flow curves for the shear-rate value of 100 $s^{-1}$, in function of time, Figure 2b, showed that in the presence of oxacillin, the viscosity of the culture almost doubled when compared with the viscosity of the culture in the absence of antibiotic, that specifically occurred during the exponential phase, in the time range ~450–550 min. This obvious increase in viscosity may be associated with the release of cytoplasmic components to the extracellular environment due to lysis of cells, due to the bacteriolytic action of oxacillin.

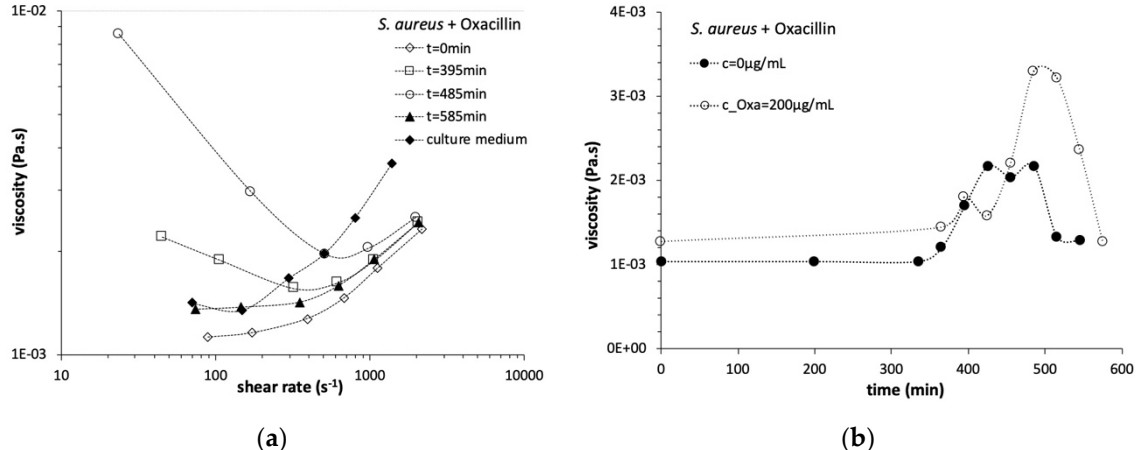

(**a**)  (**b**)

**Figure 2.** Steady-state shear flow characterization for bacterial cultures of *S. aureus*, with and without the addition of antibiotic during bacterial growth in the incubator at t = 350 min. Approach A: (**a**) flow curves of culture aliquots, removed from the incubator at specific time points during culture growth at: t = 0 min (open diamond symbol), t = 395 min (open square symbol), t = 485 min (open circle symbol), t = 585 min (full triangle symbol), culture medium (full black diamond symbol) and lines are guide to the eye. Similar behaviors were obtained for cultures without addition of oxacillin (representative curves). (**b**) Representation of shear viscosity values in function of time, obtained from the flow curves for the shear rate value of 100 $s^{-1}$: without antibiotic (full black symbol) and with oxacillin at c_Oxa = 200 µg/mL (full gray symbol). All measurements were performed at 20 °C to minimize cell growth during the assays.

### 3.3. Viscosity Growth Curves

The viscosity growth curves were determined for the two concentrations of oxacillin, added inside or outside of the rheometer (in the culture flask) after approximately 300 min of growth. For three consecutive 1 h separate points, cell viable counts were determined, and the cultures were observed and compared by optical microscopy. The images obtained, Figure 3, show the increase of cell density over time in the *S. aureus* culture and the effect of the presence of the antibiotic in the growth process.

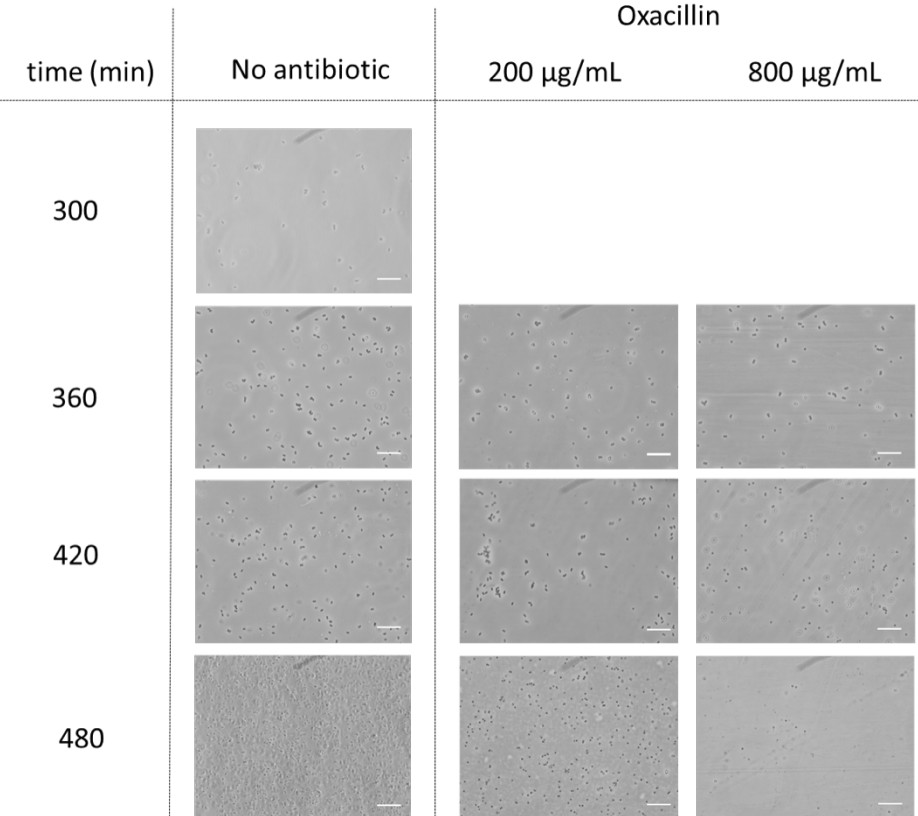

**Figure 3.** Optical microscopy images of *S. aureus* strain COL grown in the absence (left panel) and in the presence of different concentrations of oxacillin (Oxa) at 200 and 800 µg/mL (right panel), added at 300 min of growth. For each aliquot a calibrated volume of sample of 5 µL was collected and observed. An average of 10 photos were taken randomly at specific growth times: 0, 300, 360, 420 and 480 min. *S. aureus* are coccoid shaped cells with an average diameter of 0.5–1 µm. Scale bar represents 10 µm width.

For c_Oxa = 200 µg/mL, this condition did not alter the optical density profile, suggesting that the rate of cell division is maintained, see Figure 1b. However, this behavior showed no counterpart in the viability profile obtained from the progress of the CFUs/mL, see Figure 1c. Although maintaining a continually increasing pace, a decrease in the viability counts of approximately 40%, with respect to the culture with no antibiotic addition, was observed at 480 min, suggesting a slowdown in cell division. This discrepancy may be explained by a cell aggregation effect of oxacillin at sub-MIC concentrations, which was in fact observed by optical microscopy at 120 and 180 min after antibiotic addition. The principle of the CFUs measurement is that a single cell, when positioned in the agar plate will originate a single colony. When bacteria are growing in the liquid medium aggregated in groups comprising a high number of cells together, when performing the CFUs, the aggregate will be maintained in the agar plate and originate a single colony. If bacteria are growing in a non-aggregated form and instead all the cells in liquid medium are growing separately, when the medium is plated in agar to perform the CFUs, the same number of cells will originate a higher number of colonies.

For the higher concentration c_Oxa = 800 µg/mL, the expected decrease in optical density was observed at approximately 120 min after antibiotic addition, see Figure 1b, meaning a drop in cell division rate. More clearly, from the viability profile, cell division is shown to be arrested from 420 min onwards, as shown in Figure 1c. A decrease in the viability counts of approximately 90%, with respect to the culture with no antibiotic addition, was observed for this double MIC concentration. This behavior is in accordance with a cell lysis scenario, associated with the bactericidal mode of action

of oxacillin, and is corroborated by the fact that virtually no cells can be observed by optical microscopy at 480 min.

To directly explore the influence of the antibiotic presence in the rheological behavior of *S. aureus* cultures, during growth process, two different approaches were followed, by measuring the viscosity growth curve, see Figures 4 and 5. The viscosity growth curve, at sub-inhibitory antibiotic concentration follows the previously described behavior, which was framed by a microscopic model [7]. According to this model, at different growth stages, since the density of bacteria increases, the bacterial cells rearrange themselves in aggregates forming dynamic web-structures, triggering different viscoelastic responses. In these cases, the addition of the antibiotic occurred at 300 min in the incubator, approach A, after which the culture was placed in the rheometer and the viscosity growth curve measurement initiated. The cultures of *S. aureus*, with and without the addition of antibiotic, showed an almost immediate sharp increase in the viscosity, as previously reported [6] for the last case, since the exponential phase of growth initiates by this time, see Figure 4. Comparable maximum values of the viscosity were attained by each culture, corresponding to ~30× the initial value, moreover the decay in the viscosity occurred in an abrupt way and happened in ~300 min, when no antibiotic was present in the culture, while in the culture with oxacillin the viscosity decreased slowly and took almost three times as long to recover the initial viscosity value.

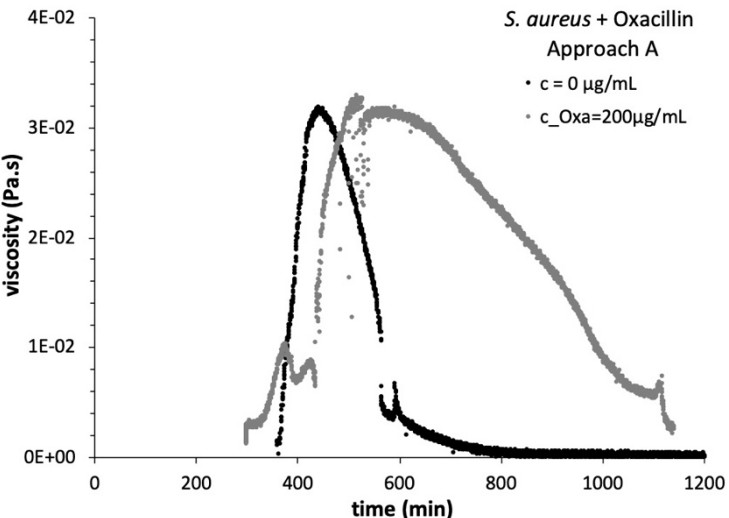

**Figure 4.** Viscosity growth curves for bacterial cultures of *S. aureus*, with and without the addition of antibiotic in the incubator at t = 300 min, approach A, followed by rheologic measurement: with no addition of antibiotic c = 0 μg/mL (full black symbol) and with oxacillin in concentration c_Oxa = 200 μg/mL (full gray symbol); all measurements were performed at a constant shear rate of 10 s$^{-1}$ (representative curves) and at 37 °C.

In Figure 5, the viscosity growth curve was measured starting with the culture of *S. aureus* and adding the antibiotic in each respective concentration, directly to the culture in the rheometer, during the assay, in the beginning of the exponential phase, approach B. For the lower antibiotic concentration, a similar behavior to the one verified with the previous addition procedure, approach A (see Figure 4), was obtained, with slight differences. The viscosity growth curves obtained with the addition of the highest antibiotic concentration showed the incapacity of the cultures to grow in such conditions, as anticipated.

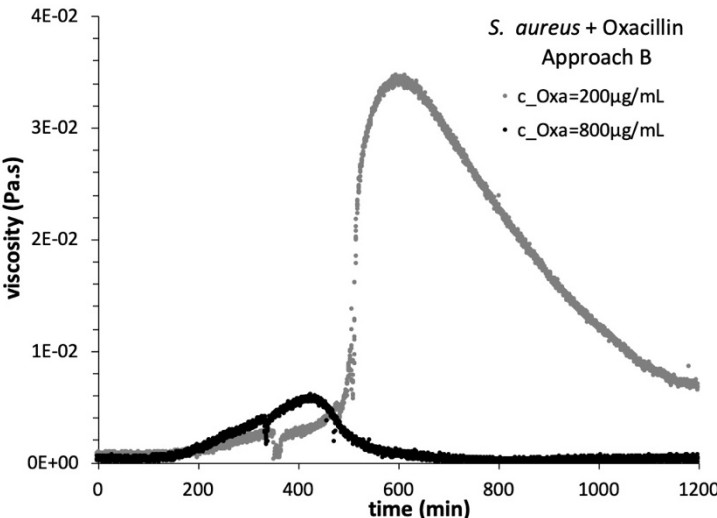

**Figure 5.** Viscosity growth curves for bacterial cultures of *S. aureus*, with the addition of the antibiotic oxacillin directly to the sample in the rheometer, approach B. The bacteriolytic antibiotic oxacillin was added at t = ~340 min for two concentrations: c_Oxa = 200 μg/mL (full gray symbol) and c_Oxa = 800 μg/mL (full black symbol); all measurements were performed at a constant shear rate of $10\ s^{-1}$ (representative curves) and at 37 °C.

## 4. Conclusions

In this study we addressed the impact of oxacillin, a β-lactam antibiotic, in the viscosity profile of the methicillin resistant *S. aureus* (MRSA) strain COL, through in-situ rheology applied during culture growth.

The addition of oxacillin, to the *S. aureus* cultures, at sub-inhibitory concentrations induced a more pronounced increase in the viscosity values during the exponential phase of growth. Several mechanisms may be contributing to this behavior, including the observed cell aggregation, partial cell lysis and the release of cell content. Finally, another hypothesis related to the secretion of compounds to the extracellular environment occurring when the bacterial cells are challenged with the antibiotic, cannot be ruled out at this moment.

For the inhibitory concentration, as expected, the cell division was rapidly arrested and as a result the rheological behaviors were drastically affected.

As future work, we also aim to explore bacterial growth in different environments and challenges with other antibiotic classes.

**Author Contributions:** Conceptualization, R.P., P.L.A., R.G.S. and C.R.L.; methodology, R.P., F.V., P.L.A., R.G.S. and C.R.L.; writing—original draft preparation, C.R.L.; writing—review and editing, R.P., P.L.A., R.G.S. and C.R.L. All authors have read and agreed to the published version of the manuscript.

**Funding:** This work was supported by FEDER through COMPETE 2020; FCT, Portuguese Foundation for Science and Technology Projects No. UID/CTM/50025/2019 (CENIMAT) and PTDC/FIS-NAN/0117/2014 (P.L.A.); PTDC/BIA-MIC/31645/2017 (R.G.S.); and the Applied Molecular Biosciences Unit, UCIBIO, which is financed by national funds from FCT (UIDB/ 04378/2020) and co-financed by the ERDF under the PT2020 Partnership Agreement No. POCI-01-0145-FEDER-007728.

**Conflicts of Interest:** The authors declare no conflict of interest.

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
