# Peer review of "Antibiotic Activity Screened by the Rheology of S. aureus Cultures"

_fluids, doi:10.3390/fluids5020076_

Round 1
Reviewer 1 Report
This is a very interesting paper which uses rheology to address the effect of the addition of oxacillin, a b-lactam antibiotic which is claim to be a well-known bacteriolytic antibiotic, in the viscosity profile of the methicillin resistant S. aureus (MRSA ) strain COL during culture growth. Some interesting conclusions were extracted, among others that the viscosity behavior can be understood in terms of different biological mechanisms also contrasted by optical microscopy: cell aggregation, cell lysis ... Therefore, the rheological discussion allowed to analyze the bacteria response to the antibiotic.
I'll just add a few comments:
First of all, I would like to highlight the difficulty in characterizing samples with such a low viscosity and a complex behavior. In fact, in Figure 2a, the viscosity curves obtained in a shear rate range between aprox 30 and 2000 s-1 were observed. Two clearly differentiated areas are distinguished in all samples. First zone is quasi Newtonian in the culture medium, 395 min, 545 min and pseudoplastic in 485 min. Second zone corresponds to thickening behavior starting from 300 s-1 in all the samples.
The shear thickening seems to be characteristic of the chosen culture medium and therefore it seems not relevant in the discussion of these data. On the contrary, the effect of antibiotic was noticed in the the Newtonian or peudoplastic zone. Precissely, it is the viscosity value taken at 100 s-1 which is compared in Figure 2b., I think that the newtonian viscosity should be a more important parameter but I asume that it can not observed for all the samples. In any case, given the complexity for the profile of these figure, I wonder if some additional comment about the reproducibility of the data could be included
On the other hand, the authors claimed that in figure 2b: in the presence of the oxacillin, the viscosity of the culture almost duplicated when compared with the viscosity of the culture in the absence of antibiotic. I wonder if any comment could be included regarding the range of time for the referred effect.
Finally, I would simply point out that it would be good to explain the abbreviation OD620 nm (It appears explained only in the legend of figure 1, but not in the text, or at least I have not seen it). Also, I wonder if a description is required for the SCC casette abbreviation.“ the mobile genetic element”
Reviewer 2 Report
This work focus on studying the antibiotic effect on bacteria using rheology (with and without antibiotic) at different concentrations regarding the inhibition of the bacteria growth at different rates as first approach. As second approach, measuring rheology at a constant rate 10s-1 in situ with adding a specific concentration of the antibiotic.
I found that the message of the work is not clear and the different experimental approaches used do not have a connection between each other and for the results rationalization. Here are my comments:
- All the figures are very poor! Please improve!
- What is this MIC concentration? it is not defined at all in the ms.
- Figure 1 – c: you report the colony growth at specific times after adding the Oxacillin.
Is the growth monitored after adding the antibiotic at 180 min from the growth (Fig1-a) or at 300 min from the growth (Fig1-b)? Because the growth after antibiotic addition seems to depend on when exactly is added after the beginning of the bacteria growth.
Rheology:
-Methods section:
In general your data is gathered during long times, did make any precaution against evaporation?
For you approach B:
- you used a parallel plate geometry in a relatively high nonlinear experiment (steady shear rate = 10 s-1). Can you argument why did you change the geometry of the approach A? You know that the rate is not homogeneous across the gap which may question your results and the gap is very big for such king of experiments. Are you sure that you data is slip-free?
You report that you were careful to avoid contact with the rotating geometry. I am sorry but this is not the case: We can clearly see that there was a contact with the rotating geometry: see the jump in the data around 340 min!
*** In addition, how can you be sure that the antibiotic is reaching all the volume of the sample and that it is not only an edge effect?
-Figure 2:
-Do you really believe data at a shear rate of 1000 s-1 and higher? This high rates are commonly accompanied with sample ejection and other instabilities for such low viscous samples!
-For a) You reported in the method section that “Each step was acquired assuring that a minimum of 1500 units of deformation was imposed to the sample” which corresponds to a time from 15 s (at 100s^-1) to 0.75 s (at 2000 s^-1).
- How the time response of the motor compares with the shear duration at each rate? It is a stress controlled, so I guess it is slower for the fast rate?
- Are you sure that you reached the steady state for each rate? I evaluated the time needed to ensure 1500 SU at a rate of 10 s-1, I found a shear duration of 150 s. However, by looking at your data in Figure 4 (still approach A) at a rate of 10s-1, the time needed to reach steady state is 800 MIN and higher !!!
-For b) It is not clear which time is plotted in the x-axis: experimental time or time elapsed after adding the antibiotic for each concentration??
-Figure 3: There is no scale bar. In general, I didn’t get what is the message from this microscopy photos ? Could you please clarify!
Reviewer 3 Report
The authors Portela et al have investigated the shear response of the MRSA bacterial response in the presence and absence of antibiotic oxacillin. The authors have attempted injecting bacteria and studying the effect of antibiotics with two specific methods i) injecting antibiotics a culture flask and ii) injecting antibiotics in-situ in the rheometer. I am not recommending publication of this manuscript because the quality of the manuscript is very poor. There is no explanation why there is a viscosity peak and then decrease for bacterial colonies and when exposed to antibiotics. There is no comparison in terms of the external exposed colonies and insitu exposed colonies. The manuscript has been written in a very substandard level in terms of language. Although there is an objective for this manuscript the experiments supporting this objective is sparse, there is a huge amount of hand waving explanations which are not convincing enough.
Line 36 “Multidrug resistance is the root of antibiotic crisis.” The first half of the sentence does not make sense. Multi drug resistance by humans or animals or cells?
Some abbreviations are introduced without expanding on them like
Line 24 sub MIC, Line 51 TGN is abbreviation for what complex medium? And Line 54 what is CFS?
Line 60 ‘stationary shear flow’ is contradictory instead please use steady shear flow which makes more sense.
Line 90 The physiological values discussed in ref 17 are 1-10 Pa and 10-60 Pa. The value of shear rates chosen in this manuscript are arbitrary and does not correspond to the claim made in Line 90. A basic calculation, stress = viscosity times shear rate
Taking values from Figure 2b Stress = 0.0015 Pas 100 s-1 = 0.15 Pa. The values of shear rate in Figs 4 and 5 is 10s-1 which is further away from physiological values of stress suggested in ref 17.
Line 131 It is not clear why the authors are performing a shear rate experiment on a stress-controlled rheometer when they can directly apply a stress of 1-10 Pa and observe the effects of viscosity on the MRSA colonies. It is advisable not to perform shear rate experiments with a stress-controlled rheometer.
Figs 1a) and b) has the same message it is not clear to me why the authors have to show both together?
Line 174 Even the culture medium itself shows shear thickening behaviour why?
Line 177- 180 The authors claims are absolutely wrong. In most of the cases the viscosity of the culture medium is higher than the MRSA in culture medium. At high shear rates the MRSA viscosity curves does not tend to culture medium viscosity curves. At the high shear rates the slopes of the viscosity are the only observation here. Also why is the viscosity of culture medium higher than the MRSA in culture medium?
Line 183 ‘almost duplicated’ what does that mean?
Figure 2a) Why are the viscosity curves performed only for a decade. In rheology viscosity curves are performed for a minimum of two to three decades. Why have the authors not performed shear rate sweep for lower shear rates(<10s-1)?
The authors should specifically note that there is an uncanny similarity of Fig 2a) with that of an earlier published result by the same authors fig 2 (ref 5). This is self-plagiarism.
Line 210 How does a decrease in viability (or some discrepancy) relate to cell aggregation is not clear?
Figure 3 barely explained in text. Why is image at 480 min for no antibiotics different from the rest of the images. It is not a clear image
Line 236 There is no mention of why there is a decrease of the viscosity with time to its original value anywhere in the text. Is it because the MRSA col accommodates to the stress applied? The authors need to make an attempt to understand this.
Line 240 It is not clear how Approach B is achieved. How is proper mixing of the antibiotic ensured with the insitu sample. How is “minimal interference with upper geometery” obtained as mentioned in the procedure. Please shoe schematic
In general, how repeatable are the experiments?
Reviewer 4 Report
The present work shows the impact of one of the most clinically relevant class of antibiotics on S. aureus bacterial cultures, and reveals distintive characteristics via steady-state shear rheology. I have found the manuscript reading quite easy, although there a few discrepancies between the main text and the graphs that I would like to highlight. For instance, Figure 1 a and b shows an arrest in growth of bacteria cells after 250 min and >400 min whereas the authors write about approximately 100 min after antibiotic addition without recalling which Figure to refer to. My thoughts went to Figure 1b (antibiotics added at 300 min). Furthermore, in Figure 2a, unless the legend is misplaced or something, all of the data with antibiotics are laying under (and not above) the culture medium at most of the applied shear rates. Exceptions are 395 min and 485 min cases at small rates. To this end, since these systems show very low viscosities in the range of mPa*s, it is a little hard to address shear thinning/thickening behavior unanimously without error bars of standard error for example. Also, the authors should improve a little bit more the discussion of their rheology data thorough the manuscript to make it more appealing even to no-bacteria experts:- Is there any reason why viscosities are lower for bacteria cell cultures with antibiotics compared to no-antibiotics cultures at most of the applied shear rates seen in Figure 2a (authors claim similar behavior is observed with/without antibiotics), and higher over time in Figure 2b? Address the role of applied shear rate.
- What is the reason of such sharp viscosity increase and different decay time at constant shear rate observed in both Figure 4 and 5? If it is related to cell lysis due to antibiotics, I do not understand the same peak value for with/without antibiotics case in Figure 4.
- It is not straightforward to me why the addition of 200 um/mL antibiotics outside and inside the rheometer lead to 30 times and only 5 times viscosity increase, respectively. How was addressed homogeneous dispersion of the antibiotics once injected?
- OD620nm not defined as optical density in the main text (only in the caption of Figure 1);
- What is the reason to use a cone-plate geometry for building flow curves and a parallel-plate geometry for constant shear experiments?
- I recommend to specify the addition time of the antibiotics in graphs textbox also for ease comparison since antibiotics addition time scale is a key parameter in the observed response;
- Mention of Figure 4 is missing in the main text between 230 and 237 lines.
Round 2
Reviewer 1 Report
I believe that the current manuscript has been greatly improved, as the authors have satisfactorily responded to the different questions and suggestions and made the necessary changes. Therefore I consider that the paper is sufficient for publication in its present form.
Author Response
We acknowledge the support of the Referee.
Reviewer 2 Report
I thank the authors for their detailed answers. I have some minor albeit important comments:
2.7: You made the choice not to make any change in the text regarding this point. I really think that the details you reported in the answer are more important than the schematic. Do you have pictures for the procedure described or reference?
2.8: The same as in my previous comment, you may have to consider to add a sentence regarding this issue because your data above 1000 s^-1 could be understood as an artifact.
2.9: You answered that “…it was possible to visualize the acquisition of the step and observe that in each one, the steady-state level was attained, where the final acquisition was made...”. Although my question which concerns the time of the stabilization of the rate was not answered, I guess that the steady state level in your answer refers to the shear rate steady state?
2.10: Thank you for you answer. However, it does not tell me whether the steady state viscosity is reached or not for flow curves presented in Fig.2. At each point (shear rate) from your flow curve (figure 2b), the viscosity of the system undergoes a transient response before reaching a steady state. This exactly what is showing in your figure 4 and 5.
I rephrase my question: Figure 4 and 5 show that the steady state is approached starting from 600 min and above (at 800 min your data is more in steady state) at a rate of 10 s^-1. This implies a total deformation of 6000 su to 8000 su which are very large against the minimum of 1500 su you choose for flow curves in Fig.2.
Again, did you check that the time used for all shear rates allows the viscosity to reach steady state in your Fig.2 ? May be this what you meant by your answer in 2.9 : “As indicated, each point was acquired guaranteeing a minimum of 1500 ud, however, in general, the application
of the shear rate at each point occurs in a much longer period of time than necessary.” ? Please refer to my comment on 2.9 above!
Reviewer 3 Report
The authors Portela et al have not been able to satisfactorily address the concerns in the manuscript. In general the manuscript still appears weak and with many flaws specifically in experimental design and data interpretation.
The approach B presented by the authors is not convincing. The authors should show the images for the dye dispersion to prove there is no peripheral accumulation. Till then it is very difficult to be convinced.
The authors will need to do more careful experiments to make this manuscript publication quality.
Unfortunately this manuscript cannot be recommended for publication.
Author Response
We have no comment to the Referee.
Reviewer 4 Report
The new version of the manuscript has shown great improvements since the first submission. The description of the results in both main text and graphs is clearer and more cohesive with Authors' previous results.
About sample loading condition check, Authors' effort was highly appreciated and their findings make data look more solid. To this end, I think it is important to mention it in the main text.
Nevertheless, I do not fully agree with Authors' decision on not including their hypothesis to describe the viscoelasticity of viscosity growth curves of S. aureus suspensions shown in Fig. 4 and 5. My observation is based on the fact that some of them can be supported by optical microscopy and stimulate further discussion and case-studies on performing rheo-optic measurements on these systems as well.
